# Efficacy of Remdesivir-Containing Therapy in Hospitalized COVID-19 Patients: A Prospective Clinical Experience

**DOI:** 10.3390/jcm10173784

**Published:** 2021-08-24

**Authors:** Alessandro Russo, Erica Binetti, Cristian Borrazzo, Elio Gentilini Cacciola, Luigi Battistini, Giancarlo Ceccarelli, Claudio Maria Mastroianni, Gabriella d’Ettorre

**Affiliations:** 1Infectious and Tropical Disease Unit, Department of Medical and Surgical Sciences, “Magna Graecia” University of Catanzaro, 88100 Rome, Italy; 2Policlinico “Umberto I”, Department of Public Health and Infectious Diseases, “Sapienza” University of Rome, 00185 Rome, Italy; erica.binetti@uniroma1.it (E.B.); cristian.borrazzo@uniroma1.it (C.B.); Gentilini.1979701@studenti.uniroma1.it (E.G.C.); battistini.1586873@studenti.uniroma1.it (L.B.); giancarlo.ceccarelli@uniroma1.it (G.C.); claudio.mastroianni@uniroma1.it (C.M.M.); gabriella.dettorre@uniroma1.it (G.d.)

**Keywords:** remdesivir, COVID-19, pneumonia, non-invasive ventilation, mechanical ventilation

## Abstract

Objectives: Remdesivir is currently approved for the treatment of COVID-19. The recommendation for using remdesivir in patients with COVID-19 was based on the in vitro and in vivo activity of this drug against SARS-CoV-2. Methods: This was a prospective observational study conducted on a population of patients hospitalized for COVID-19. The primary endpoint of this study was the impact of remdesivir-containing therapy on 30-day mortality; the secondary endpoint was the impact of remdesivir-containing therapy on the need for high-flow oxygen therapy (HFNC), non-invasive ventilation (NIV), or mechanical ventilation. The data were analyzed after propensity score matching. Results: A total of 407 patients with SARS-CoV-2 pneumonia were consecutively enrolled. Out of these, 294 (72.2%) were treated with remdesivir and 113 (27.8%) were not. Overall, 61 patients (14.9%) were treated during hospitalization with HFNC, NIV, or mechanical ventilation, while 30-day mortality was observed in 21 patients (5.2%). Univariate analysis of patients treated with remdesivir or not showed no differences in 30-day mortality (4% vs. 6%, *p* = 0.411) in the two study groups. Cox regression analysis, after propensity score matching, showed that therapies, including remdesivir-containing therapy, were not statistically associated with 30-day survival or mortality. The Kaplan–Meier curves of 30-day survival in patients treated with remdesivir or not before (*p* = 0.24) and after (*p* = 0.88) propensity score matching showed no differences between the two study groups. Finally, patients treated with remdesivir or not showed the same need for HFNC/NIV or mechanical ventilation. Conclusions: This real-life experience of remdesivir use in hospitalized patients with COVID-19 was not associated with significant increases in rates of survival or reduced use of HFNC/NIV or mechanical ventilation compared with patients treated with other therapies not including remdesivir.

## 1. Introduction

Corticosteroids, particularly dexamethasone, have been the standard of care in patients with severe coronavirus disease (COVID-19) since the publication of the results of the RECOVERY trial [1,2]. Other anti-inflammatory and immunomodulatory therapies, such as tocilizumab and baricitinib, can be considered in patients with severe COVID-19 [2]. Remdesivir has been the only antiviral medication suggested for the treatment of hospitalized patients with severe COVID-19 [2].

Remdesivir is an inhibitor of viral RNA polymerase that was initially studied on Ebola virus [3]. Remdesivir was found to have in vitro and in vivo activity against severe acute respiratory syndrome coronavirus 2 (SARS-CoV-2) and has an acceptable safety profile [4,5,6]. Therefore, remdesivir was studied in large clinical trials that were initiated in the early phase of the pandemic [1,2,3,4,5,6,7]. In particular, a large trial conducted by the ACTT-1 study group revealed that remdesivir was superior to placebo in shortening the time to recovery in hospitalized patients with COVID-19 [4]. The results of this study led to approval by [8,9] for the treatment of patients with COVID-19. A subsequent meta-analysis of randomized trials conducted by Kaka et al. demonstrated that remdesivir may reduce mortality in patients that require supplemental oxygen but are not on mechanical ventilation [10].

Real-world data on the efficacy of remdesivir in the treatment of hospitalized patients with COVID-19 are needed. Therefore, we performed this prospective observational study aiming to investigate the impact of remdesivir on 30-day mortality and the need for invasive and non-invasive ventilation in a large Italian institution.

## 2. Materials and Methods

### 2.1. Study Design and Data Collection

This prospective observational study included patients admitted to Policlinico Umberto I of the University Hospital of Rome, Italy, from October 2020 to February 2021. Inclusion criteria were (1) positive SARS-CoV-2 real-time polymerase chain reaction test or antigenic test on nasopharyngeal swab, (2) pneumonia diagnosed either by thorax CT or chest x-ray, and (3) need for hospitalization. Patients who required high-flow oxygen therapy (HFNC) or non-invasive ventilation (NIV) or mechanical ventilation at the time of hospitalization were excluded from this analysis.

All patients were evaluated in a dedicated emergency department by dedicated infectious diseases specialists who identified patients with SARS-CoV-2 pneumonia, followed the patients during hospitalization, and collected all data prospectively without interfering with patient management. This observational study was conducted according to the principles stated in the Declaration of Helsinki, and it conforms to standards currently applied in our country. This study was approved by the local EC. Informed consent was obtained from the patients.

Data were extracted from the hospital’s computerized databases and the patients’ medical records. The following data were collected: demographics, clinical and laboratory findings, comorbidities, Charlson comorbidity index, microbiologic data, date of COVID-19 diagnosis, radiological characteristics of the pneumonia, therapies used, concomitant infections, duration of mechanical ventilation, time of negative nasopharyngeal swab, need for oxygen or ventilation support during the hospital stay, length of ICU stay, and length of hospital stay. Development of moderate to severe ARDS was defined as the acute onset of hypoxemia, manifestations of pneumonia of noncardiac origin on chest computed tomography imaging, and a PaO_2_/FiO^2^ ratio of less than 200 mmHg according to the Berlin definition [11].

Remdesivir was administered, after written informed consent was obtained, to patients according to the following criteria: presence of pneumonia, need for low-flow oxygen therapy, less than 10 days from the onset of symptoms, no need for HFNC or NIV or mechanical ventilation, alanine aminotransferase no more than 5-fold the upper limit of the reference range, and estimated glomerular filtration rate (eGFR) greater than 30 mL/min. A 5 day regimen was prescribed in all cases. Patients without these criteria were not eligible for remdesivir treatment.

All patients were followed until discharge or death. All discharged patients were followed for 30-days to assess outcomes.

### 2.2. Endpoints and Statistical Analysis

The primary endpoint of this study was the impact of remdesivir-containing therapy on 30-day mortality in hospitalized patients with SARS-CoV2 pneumonia. The secondary endpoint was the impact of remdesivir-containing therapy on the need for NIV or mechanical ventilation.

To reduce the impact of treatment selection bias in the estimation of treatment effects, propensity score matching was conducted with the nearest neighbor matching procedure without replacement [12]. Variables were selected for inclusion in the propensity score based on the potential impact on receipt of remdesivir and the association with mortality [13]. The variables included were steroids, antibiotics (excluding macrolides), age, gender, oxygen, comorbidities, CRP concentrations and the use of LMWH during hospital admission. A propensity score density plot and a Love plot were generated to examine the balance of propensity score and covariate distribution between the two groups (see Appendix A).

To evaluate the demographic factors, Welch’s t-test assuming unequal variances was used for continuous independent variables, while Pearson’s Chi-square or Fisher’s exact test was used, where appropriate, for categorical variables. Welch’s analysis of variance (ANOVA) was used to assess group differences for continuous outcomes. Welch’s t-test assuming unequal variances was used for post hoc comparisons.

All tests were two tailed, and a *p*-value < 0.05 was considered statistically significant. Results are expressed as the mean with standard deviation (±SD) for continuous normally distributed variables and as a count (*n*) and percentage (%) for categorical variables. Multivariate analysis was used to identify independent predictors of 30-day mortality and the need for NIV or mechanical ventilation. Matched bivariate analysis was conducted using a conditional logistic regression model, incorporating all variables found to be significant in the univariate analysis (*p* < 0.05) with a stepwise method. Matched multivariate models were constructed using Cox proportional hazard (HR) regression if appropriate, accounting for clustering of matched pairs. The final selected model was tested for confounding. In addition, a 95% confidence interval was calculated for HR. Survival was analyzed by Kaplan–Meier curves. All data were analyzed using a commercially available statistical software package (SPSS Statistics for Mac, 22.0; IBM Corp., Armonk, NY, USA).

## 3. Results

During the study period, 407 patients with SARS-CoV-2 pneumonia were consecutively enrolled. Out of these, 294 (72.2%) were treated with remdesivir and or 113 (27.8%) were not (control group). The mean time for remdesivir administration was 5.2 days (±2.9) from the onset of symptoms. Overall, 61 patients (14.9%) were treated during hospitalization with HFNC, NIV, or mechanical ventilation, and the 30-day mortality rate was 5.2% (21 patients).

Table 1 reports the univariate analysis of demographics and clinical characteristics of COVID-19 patients treated with remdesivir or not. Statistically significant differences were observed in the remdesivir group with regard to male sex (80% vs. 62%, *p* < 0.001), fever (79% vs. 50%, *p* < 0.001), cough (50% vs. 29%, *p* < 0.001), and dyspnea (57% vs. 37%, *p* < 0.001) compared to patients in the control group. No statistically significant differences were observed in the remdesivir group with regard to age (63.2 vs. 62.5 years, *p* = 0.717), and days to negative nasopharyngeal swab (22.07 vs. 24.77 days, *p* = 0.378).

In-hospital treatments for COVID-19 patients are reported in Table 2. A comparison between patients treated with remdesivir or not shows that steroids (93% vs. 81%, *p* < 0.001) and LMWH (93% vs. 52%, *p* < 0.001) were more frequently prescribed in the remdesivir group; antibiotic therapy (58% vs. 27%, *p* < 0.001) was more frequently prescribed for patients in the control group; and no differences were reported regarding the use of HFNC/NIV or mechanical ventilation in the two study groups.

In Table 3 are reported outcomes of hospitalized patients in the two study groups. No statistically significant differences were observed about length of hospital stay (15.02 vs. 16.06 days, *p* = 0.487), bacterial co-infection (20% vs. 21%, *p* = 0.928), and 30-day mortality (4% vs. 6%, *p* = 0.411).

Appendix A reports the results of univariate analysis before and after propensity score matching to evaluate the impact of the remdesivir-containing regimen on the study population. Figure 1 shows Kaplan–Meier curves for 30-day survival of patients treated with remdesivir or not before (*p* = 0.24) and after (*p* = 0.88) propensity score matching, showing no differences between the 2 study groups. Standardized differences before and after propensity score matching are reported in Appendix A.

Multivariate Cox regression analysis of 30-day mortality after propensity score matching is reported in Table 4. Therapies, including remdesivir-containing therapy, were not statistically associated with 30-day survival or mortality. However, mechanical ventilation (HR 4.22, 95% CI 5.4–16.2, *p* = 0.003) was independently associated with 30-day mortality.

Finally, multivariate Cox regression was used to analyze the need for non-invasive or invasive ventilation after propensity score matching (see Table 5). The data show that comorbidities and therapies, including the remdesivir-containing regimen, were not independently associated with a lower or higher risk of needing HFNC/NIV or mechanical ventilation.

## 4. Discussion

This prospective clinical study reports a real-life experience with the use of remdesivir in a large population of consecutively hospitalized patients with COVID-19. Our data, also after propensity score matching, show that the remdesivir-containing regimen was not associated with 30-day patient survival compared to treatment with other therapies not including remdesivir. Moreover, the remdesivir-containing regimen was not independently related to the need for HFNC/NIV or mechanical ventilation.

In Italy, remdesivir was specifically licensed for the treatment of COVID-19 in hospitalized patients with pneumonia who require oxygen therapy but not HFNC/NIV or mechanical ventilation at the time of remdesivir prescription [14].

Different data were reported around the world regarding the efficacy of remdesivir, taking into account different outcomes. In patients with severe COVID-19, treatment with remdesivir was significantly associated with higher recovery rates and lower mortality compared to standard-of-care treatment without remdesivir [15]. In this study, the mortality rate was significantly lower for patients treated with remdesivir (7.6%) compared with control groups (12.5%). Conversely, data from the Solidarity trial, conducted in 30 countries [16], showed no decrease in in-hospital mortality in patients treated with remdesivir, with the important limitation that other outcomes (clinical improvement and adverse events) were not carefully evaluated.

Recent real-word studies reporting data on the use of remdesivir [17] also compared it with lopinavir/ritonavir [18]. Some important meta-analysis showed that COVID-19 patients receiving remdesivir showed significantly higher rates of recovery and hospital discharge with lower rates of serious adverse events when compared to patients receiving other treatments [19,20]. However, these analyses also noted that there were no significant differences in clinical improvement and rate of mortality during hospitalization. Specifically, mortality was the main outcome reported in all analyzed studies, which showed no significant decrease in mortality if they were not adequately powered for this outcome [12].

Wang et al. [21] reported the first double-blind randomized clinical trial evaluating patients with a mean interval from symptom onset to enrollment of 12 days. No differences in mortality were recorded in the two arms, and the authors highlighted a possible trend of clinical benefit in patients treated with remdesivir. Of importance, a large number of patients in this study were also treated with steroids (65% in the remdesivir arm and 68% in the placebo arm), which may have confounded the results and conclusions. A strength of our study, with the limitation of the non-randomized cohort, was weighting all possible therapeutic confounders, including the use of steroids and LMWH [22,23].

Beigel et al. [7] randomized 1062 patients hospitalized with COVID-19 and evidence of pneumonia to remdesivir or placebo. This study demonstrated that remdesivir was superior to placebo in shortening the time to recovery in COVID-19 patients, with a trend toward survival benefit at day 29, without statistically significant differences. Of interest, the authors reported a beneficial effect of remdesivir in severe COVID-19 patients who did not require mechanical ventilation at enrollment; they suggested to start remdesivir early in the disease course.

Finally, in another randomized clinical trial [24] of patients with low to moderate COVID-19 (no oxygen requirement, but about 15% of patients required oxygen at the time of enrollment), the authors randomized 596 patients in a 1:1:1 ratio to receive a 5 day or 10 day course of remdesivir or standard-of-care therapy. The 5 day, but not the 10 day treatment showed a statistically significant difference with regard to the main clinical outcome. In the analysis, excluding patients who required oxygen at baseline, statistically significant differences favoring remdesivir over standard care were reported.

Our study has some limitations. First, considering the monocentric design, these results might be affected by local practice in the management of COVID-19. Second, although the criteria for HFNC/NIV and mechanical ventilation were based on the degree of respiratory impairment, critically ill elderly patients with ultimately fatal diseases were probably excluded from non-invasive/invasive ventilation, modifying the interpretation of some interventions; moreover, the small sample size did not permit definitive conclusions, including about HFNC/NIV and mechanical ventilation (only 61 patients were analyzed) and some important variables (like body mass index [BMI]) were not available for all study population. Third, this analysis evaluated consecutively hospitalized patients independently from COVID-19 severity, as demonstrated by the low mortality rate (6% of remdesivir group vs. 4% of those not treated with remdesivir). Finally, the analysis of the beneficial effects of treatments should be interpreted cautiously because it was not conducted with randomized groups and might therefore be affected by several measured and unmeasured confounding factors. However, the comparison of patients treated and not treated with remdesivir was based on a robust statistical methodology appropriate for non-randomized cohort studies about therapy.

## 5. Conclusions

In conclusion, in this real-life experience, the use of remdesivir in hospitalized patients with COVID-19 was not associated with significantly increased rates of survival or reduced use of HFNC/NIV or mechanical ventilation compared to treatment with other therapies not including remdesivir. These results suggest the need to conduct other RCTs to evaluate the impact of remdesivir in hospitalized COVID-19 patients at different stages of the disease or in combination with other drugs [5]. However, considering its safety profile and the lack of alternative drugs, remdesivir should continue to be administered for patients with COVID-19.

## Figures and Tables

**Figure 1 jcm-10-03784-f001:**
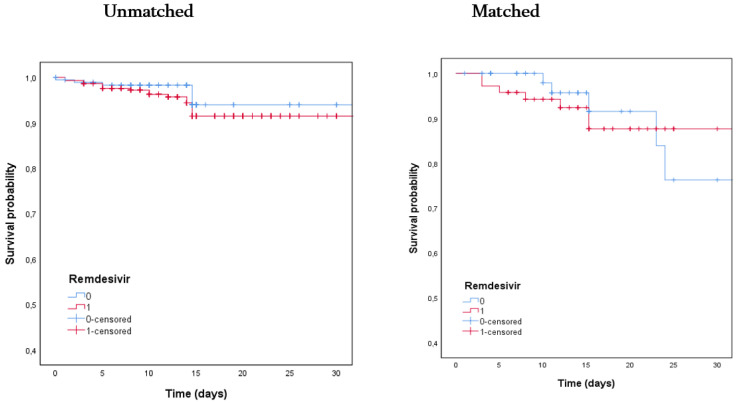
Kaplan–Meier curves for 30-day survival of patients treated with remdesivir (red line) or not (blue line) before (*p* = 0.24) and after (*p* = 0.88) propensity score matching.

**Table 1 jcm-10-03784-t001:** Univariate analysis regarding demographics and clinical characteristics of COVID-19 patients treated with remdesivir or not.

Variable	Control Group *n* = 113 (%)	Remdesivir *n* = 294 (%)	*p*-Value
Male sex	70 (62%)	250 (80%)	<0.001
Age, years, mean (± SD)	62.5 (±20)	63.2 (±15.3)	0.717
Days from symptoms/positive nasopharyngeal swab to admission, mean (± SD)	4.5 (±4.3)	5.3 (±3.8)	0.084
Charlson comorbidity index (± SD)	2.5 (±2.1)	2.6 (±1.9)	0.719
Cardiovascular disease	17 (15%)	33 (11%)	0.203
COPD	19 (17%)	31 (10%)	0.051
Chronic kidney disease	10 (9%)	18 (6%)	0.256
Liver cirrhosis	2 (2%)	12 (4%)	0.293
Diabetes mellitus	21 (19%)	53 (17%)	0.672
Solid lung cancer (primary or metastasis)	1 (1%)	3 (1%)	1
Fever	56 (50%)	246 (79%)	<0.001
Cough	33 (29%)	157 (50%)	<0.001
Dyspnea	42 (37%)	178 (57%)	<0.001
Gastrointestinal symptoms (diarrhea, abdominal discomfort, nausea, vomiting)	16 (14%)	58 (18%)	0.271
Fatigue	21 (19%)	57 (19%)	1
Arthralgia/myalgia	13 (12%)	45 (14%)	0.46
Anosmia	3 (3%)	9 (3%)	1
Conjunctivitis	0 (0%)	2 (1%)	0.397
Chest pain	5 (4%)	11 (3%)	0.655
Parenchymal thickening	72 (64%)	232 (74%)	0.046
Interstitial lung disease	16 (14%)	15 (5%)	<0.001
Pleural effusion	20 (18%)	26 (9%)	0.012
Bronchiectasis/emphysema	27 (24%)	50 (15%)	0.032
White blood cells ×10^3^/uL, mean (±SD)	7.38 (±3.59)	8.06 (±5.99)	0.287
Neutrophils ×10^3^/uL, mean (±SD)	5.60 (±3.61)	6.31 (±5.31)	0.211
Lymphocytes ×10^3^/uL, mean (±SD)	1.18 (±0.65)	1.12 (±2.28)	0.769
Platelets ×10^3^/uL, mean (±SD)	247.66 (±100.85)	218.47 (±81.73)	0.004
D-dimer ng/mL, mean (±SD)	1365.71 (±1456.18)	814.91 (±766.45)	<0.001
Ferritin ng/mL, mean (±SD)	692.35 (±942.17)	645.02 (±489.52)	0.591
Procalcitonin ng/mL, mean (±SD)	1.20 (±6.4)	0.61 (±3.73)	0.334
LDH mU/mL, mean (±SD)	288.46 (±163.73)	302.37 (±119.57)	0.3
CPK U/L, mean (±SD)	278.90 (±1430.63)	149.47 (±161.35)	0.17
Lactates mmol/L, mean (±SD)	1.57 (±0.36)	1.33 (±0.83)	0.328
C-reactive protein mg/dL, mean (±SD)	4.91 (±6.61)	8.23 (±21.13)	0.116
PaO_2_/FiO_2_, mean (±SD)	315.67 (±117.14)	329.94 (±98.36)	0.411
Aspartate transaminase U/L, mean (±SD)	34.16 (±47.76)	35.24 (±27.41)	0.784
Alanine transaminase U/L, mean (±SD)	32.20 (±28.42)	33.24 (±28.84)	0.755

Legend. SD, standard deviation; COPD, chronic obstructive pulmonary disease; LDH, lactate dehydrogenase; CPK, creatine phosphokinase.

**Table 2 jcm-10-03784-t002:** In-hospital treatments for COVID-19 patients treated with remdesivir or not.

Variable	No Remdesivir *n* = 113 (%)	Remdesivir *n* = 294 (%)	*p*-Value
Steroids	92 (81%)	289 (93%)	<0.001
Antibiotics (excluding macrolides)	65 (58%)	83 (27%)	<0.001
Macrolides	74 (65%)	146 (46%)	<0.001
Low-molecular-weight heparin	59 (52%)	280 (93%)	<0.001
No need for oxygen therapy	24 (21.2%)	-	<0.001
Low-flow oxygen therapy	65 (57.5%)	275 (87%)	<0.001
HFNC/NIV	17 (15%)	33 (10%)	0.155
Mechanical ventilation	1 (1%)	10 (3%)	0.2

Legend. HFNC, high-flow nasal cannula; NIV, non-invasive ventilation.

**Table 3 jcm-10-03784-t003:** Outcomes of COVID-19 patients treated with remdesivir or not.

Variable	No Remdesivir *n* = 113 (%)	Remdesivir *n* = 294 (%)	*p*-Value
Steroids	92 (81%)	289 (93%)	<0.001
Antibiotics (excluding macrolides)	65 (58%)	83 (27%)	<0.001
Macrolides	74 (65%)	146 (46%)	<0.001
Low-molecular-weight heparin	59 (52%)	280 (93%)	<0.001
No need for oxygen therapy	24 (21.2%)	-	<0.001
Low-flow oxygen therapy	65 (57.5%)	275 (87%)	<0.001
HFNC/NIV	17 (15%)	33 (10%)	0.155
Mechanical ventilation	1 (1%)	10 (3%)	0.2

**Table 4 jcm-10-03784-t004:** Multivariate Cox regression analysis of 30-day mortality after propensity score matching.

Variable	HR	95% CI	*p*-Value
Charlson comorbidity index < 2 points	0.2	0.1–2.0	0.012
Chronic kidney disease	1.8	0.1–140.2	0.812
COPD	3	0.08–95	0.523
Bacterial co-infection	0.88	0.1–7.81	0.772
Low-molecular-weight heparin	0.2	0.013–2.2	0.174
Macrolides	23.3	0.273–20.8	0.17
Antibiotics (excluding macrolides)	1.54	0.18–13	0.822
Steroids	0.12	0.0–1.24	0.892
No need for oxygen therapy	2.12	0.234–8.6	0.782
Low-flow oxygen therapy	10.7	0.434–176.4	0.122
Remdesivir	0.87	0.12–1.2	0.184
HFNC/NIV	182	0.954–336.7	0.054
Mechanical ventilation	4.22	5.4–16.2	0.003

Legend. COPD, chronic obstructive pulmonary disease; HFNC, high-flow nasal cannula; NIV, non-invasive ventilation.

**Table 5 jcm-10-03784-t005:** Multivariate Cox regression analysis of need for non-invasive or invasive ventilation after propensity score matching.

Variable	HR	IC	*p*-Value
Charlson comorbidity index <2 points	0.2	0.1–1.4	0.156
Chronic kidney disease	1.8	0.24–4.2	0.788
COPD	1.03	0.068–15.64	0.548
Bacterial co-infection	0.7	0.1–1.8	0.768
Low-molecular-weight heparin	0.18	0.01–2.2	0.174
Macrolides	0.4	0.01–8.8	0.494
Antibiotics (excluding macrolides)	1.6	0.18–12.67	0.722
Steroids	3.6	0.8–137	0.892
Remdesivir	1.03	0.8–15.6	0.802

Legend. COPD, chronic obstructive pulmonary disease.

## Data Availability

The data are available upon request by email to a.russo@unicz.it.

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
