# Peer review of "Efficacy of Remdesivir-Containing Therapy in Hospitalized COVID-19 Patients: A Prospective Clinical Experience"

_jcm, 2021, doi:10.3390/jcm10173784_

Round 1

Reviewer 1 Report

Authors examined the efficacy of remdesivir-containing therapy in hospitalized 2 COVID-19 patients: a prospective clinical experience.

This study was a prospective, observational study conducted on a large population of patients 18 hospitalized for COVID-19. The primary endpoint of the study was to evaluate the impact on 30-day mortality; secondary endpoint was the impact of therapy on the need of high flow oxygen therapy (HFNC) or non-invasive 21 ventilation (NIV) or mechanical ventilation. Data were analyzed after propensity score matching.

Although this manuscript is interesting, several issues arise.

Major points

This result is important but negative for remdesivir-containing therapy.

The findings of CT or MRI may be helpful.

Table 1 and 2 including before and after treatments may be helpful.

HbA1C, T-cho, TG and uric acid may be helpful in Table 1.

In underlying diseases, hypertension, diabetes mellitus and hyperlipidemia may be helpful.

How did authors use COVID-19 or SARS-CoV-2?

Did authors analyze laboratory data in Table 3 or 4?

Minor points

Table 1 is too busy.

Figure-1 should be further explained in legends.

Author Response

Authors examined the efficacy of remdesivir-containing therapy in hospitalized 2 COVID-19 patients: a prospective clinical experience.

This study was a prospective, observational study conducted on a large population of patients 18 hospitalized for COVID-19. The primary endpoint of the study was to evaluate the impact on 30-day mortality; secondary endpoint was the impact of therapy on the need of high flow oxygen therapy (HFNC) or non-invasive 21 ventilation (NIV) or mechanical ventilation. Data were analyzed after propensity score matching.

Although this manuscript is interesting, several issues arise.

R: Dear reviewer, thanks for all your efforts to improve quality of our analysis and data presentation.

Major points

This result is important but negative for remdesivir-containing therapy.

R: in Discussion we reported a high number of limitations in order to highlight all the limits of this analysis. However, the strength of the study was the use of propensity score matching to balance the effect of treatments on outcomes.

The findings of CT or MRI may be helpful.

R: data about lung CT are reported in Table 1. MRI was not performed routinely in these patients.

Table 1 and 2 including before and after treatments may be helpful.

R: this is an important observation. However, we have to explain that, considering that remdesivir was administered in patients with less than 10 days from the onset of symptoms, all patients were treated or not with remdesivir from the Emergency Department. We report now in Results section that the mean time for remdesivir administration was 5.2 days (± 2.9) from the onset of symptoms.

HbA1C, T-cho, TG and uric acid may be helpful in Table 1.

R: these variables were recorded in a very small percentage of cases. Then, we decided to don’t report all these informations also to reduce the length of Table 1. As a matter of fact, the main focus of our analysis was the impact of treatments on outcome.

In underlying diseases, hypertension, diabetes mellitus and hyperlipidemia may be helpful.

R: Thanks also for this observation. Diabetes mellitus is reported in the Table 1, we included hypertension in “cardiovascular disease” while no data about hyperlipidemia were recorded.

How did authors use COVID-19 or SARS-CoV-2?

R: we modified the manuscript to use appropriately these terms.

Did authors analyze laboratory data in Table 3 or 4?

R: as reported in Methods (statistical analysis) matched multivariate models was constructed using Cox proportional hazards (HRs) regression if appropriate, accounting for clustering on matched pairs. Then, the final selected model was tested for confounding with the selection of most important variables (mainly therapeutic interventions considering the endpoint of this study). We reported in supplementary material Table 1 univariate analysis after propensity score matching to show the performance of all the variables.

Minor points

Table 1 is too busy.

R: we deleted some variables.

Figure-1 should be further explained in legends.

R: we further explained the Figure 1, as required.

Reviewer 2 Report

The paper “Efficacy of remdesivir-containing therapy in hospitalized COVID-19 patients: a prospective clinical experience” provides real-world data on the remdesivir use in patients with COVID-19. The Authors did not find an impact of RDV on the mortality and need of high flow oxygen therapy and mechanical ventilation.

Below I have presented some questions and critical comments to the manuscript. 

  1. RDV was administered within 10 days from the onset of symptoms. Could you provide a median time of administration?
  2. Please clarify how long patients were followed up. In the section Materials and Methods there is information that all patients were followed up until discharge or death. However, the median time of hospitalization was 16,06 and 15,02 days in non-RDV and RDV group, respectively. What about patients discharged after less than 30 days? Do they were followed up for 30-day mortality or only in-hospital deaths were documented?
  3. Table 2 presented the comparison between patients treated with and without RDV reporting the rates of patients on low-flow oxygen therapy and not need oxygen supplementation together. Could you provide a more detailed comparison considering separately patients without oxygen therapy and those receiving low-flow oxygen therapy? Results from a clinical trial performed by Beigel et al., which was a basis for registration of RDV in COVID-19 patients, reported a difference between patients not receiving oxygen and on low-flow oxygen therapy, so the performing of the analysis considering these two groups of patients is strongly recommended to evaluate the impact of RDV treatment.
  4. I suggest the Authors comment papers on RWE use of RDV: 1. Remdesivir-based therapy improved the recovery of patients with COVID-19 in the multicenter, real-world SARSTer study. Pol Arch Intern Med. 2021 Jan 29;131(1):103-110. doi: 10.20452/pamw.15735. Epub 2020 Dec 31. PMID: 33382547. and 2. Real-life use of remdesivir in hospitalized patients with COVID-19. Rev Esp Quimioter. 2021 Apr;34(2):136-140. doi: 10.37201/req/018.2021. Epub 2021 Mar 6. PMID: 33675220; PMCID: PMC8019465.

Author Response

The paper “Efficacy of remdesivir-containing therapy in hospitalized COVID-19 patients: a prospective clinical experience” provides real-world data on the remdesivir use in patients with COVID-19. The Authors did not find an impact of RDV on the mortality and need of high flow oxygen therapy and mechanical ventilation.

Below I have presented some questions and critical comments to the manuscript. 

R: Dear reviewer, thank you very much for these important observations to improve the quality of our analysis.

  1. RDV was administered within 10 days from the onset of symptoms. Could you provide a median time of administration?

R: this is an important observation. We reported in Results section this information.

  1. Please clarify how long patients were followed up. In the section Materials and Methods there is information that all patients were followed up until discharge or death. However, the median time of hospitalization was 16,06 and 15,02 days in non-RDV and RDV group, respectively. What about patients discharged after less than 30 days? Do they were followed up for 30-day mortality or only in-hospital deaths were documented?

R: thank you also for this observation. All patients were followed-up during hospitalization, but patients were contacted also after discharge to assess 30-day outcome. We modified Methods. 

  1. Table 2 presented the comparison between patients treated with and without RDV reporting the rates of patients on low-flow oxygen therapy and not need oxygen supplementation together. Could you provide a more detailed comparison considering separately patients without oxygen therapy and those receiving low-flow oxygen therapy? Results from a clinical trial performed by Beigel et al., which was a basis for registration of RDV in COVID-19 patients, reported a difference between patients not receiving oxygen and on low-flow oxygen therapy, so the performing of the analysis considering these two groups of patients is strongly recommended to evaluate the impact of RDV treatment.

R: we modified analysis as required and reported data in Table 2 and 3.

  1. I suggest the Authors comment papers on RWE use of RDV: 1. Remdesivir-based therapy improved the recovery of patients with COVID-19 in the multicenter, real-world SARSTer study. Pol Arch Intern Med. 2021 Jan 29;131(1):103-110. doi: 10.20452/pamw.15735. Epub 2020 Dec 31. PMID: 33382547. and 2. Real-life use of remdesivir in hospitalized patients with COVID-19. Rev Esp Quimioter. 2021 Apr;34(2):136-140. doi: 10.37201/req/018.2021. Epub 2021 Mar 6. PMID: 33675220; PMCID: PMC8019465.

R: we discussed these articles

Reviewer 3 Report

I read with great interest your Manuscript on efficacy of Remdesivir in treatment of hospitalized COVID-19 patients

This is an important question of high clinical relevance, and I applaud your efforts.

The strengths of your study are:

-clinically relevant question; this question has great consequences on public health across the word at this time, and I am glad at the initiative you took to address this question

-Primary and secondary outcomes are carefully chosen, and are clinically relevant

Major limitations:

-I had hard time understanding several sentences through out the manuscript. A few examples are:

                        Abstract line 25: “61 patients were treated with…” does this mean 61 out of total 407 that were included in the study? Based on the tables it does seem to be the case, but is very hard to comprehend this from the abstract.

                        Abstract line 25. What does global in-hospital mortality mean?

                        Abstract line 28-29: 30-day mortality is your primary outcome it is important to give mortality numbers in the two groups and p value for these numbers in abstract.

                        Abstract line 33: this line is grammatically incorrect

-Invasive mechanical ventilation is a huge risk for mortality in COVID-19 patients based on multiple studies. The Hazard ratio of death in your study is higher with HFNC/NIC compared to invasive mechanical ventilation. I believe this is small sample size issue. As is confirmed by wide confidence interval of (0.954 – 336.73) for the HFNC/NIV group. Furthermore, there were only 11 intubate patients in total (10 in remdesivir arm and 1 in control). With such small sample size, it is hard to rely on this finding.

-Similarly, there are multiple grammatical errors through out the manuscript making it hard to read. Few Examples: introduction line 52, results line 150

-It may be easier to call “no remdesivir” arm as control arm in tables and figures. Figure 1 is key figure of paper. Unclear if the solid line on Kaplan Meier curve is control or Remdesivir arm? Based on data appears that solid line is control but it should be mentioned in figure legent.

-Remdsivir was not given to patients with GFR < 30 OR ALT < 5 times upper limit normal. Which is based on current guidelines. This adds a selection bias to study. The control group may have more patients with kidney and liver disease who were not able to get Remdesivir. Can authors shed some light on how that selection bias was addressed.

Author Response

I read with great interest your Manuscript on efficacy of Remdesivir in treatment of hospitalized COVID-19 patients

This is an important question of high clinical relevance, and I applaud your efforts.

The strengths of your study are:

-clinically relevant question; this question has great consequences on public health across the word at this time, and I am glad at the initiative you took to address this question

-Primary and secondary outcomes are carefully chosen, and are clinically relevant

R: Dear reviewer, we are really grateful for your effort to improve quality of this manuscript.

Major limitations:

-I had hard time understanding several sentences through out the manuscript. A few examples are:

                        Abstract line 25: “61 patients were treated with…” does this mean 61 out of total 407 that were included in the study? Based on the tables it does seem to be the case, but is very hard to comprehend this from the abstract.

R: we modified the sentence also in Results section.

                        Abstract line 25. What does global in-hospital mortality mean?

R: we removed “global” from abstract and Results section.

                        Abstract line 28-29: 30-day mortality is your primary outcome it is important to give mortality numbers in the two groups and p value for these numbers in abstract.

R: we modified abstract as required.

                        Abstract line 33: this line is grammatically incorrect

R: we modified the sentence.

-Invasive mechanical ventilation is a huge risk for mortality in COVID-19 patients based on multiple studies. The Hazard ratio of death in your study is higher with HFNC/NIC compared to invasive mechanical ventilation. I believe this is small sample size issue. As is confirmed by wide confidence interval of (0.954 – 336.73) for the HFNC/NIV group. Furthermore, there were only 11 intubate patients in total (10 in remdesivir arm and 1 in control). With such small sample size, it is hard to rely on this finding.

R: we totally agree with your suggestion. To reduce this bias we use propensity score matching. We reported this limitation about the small sample size in Discussion.

-Similarly, there are multiple grammatical errors through out the manuscript making it hard to read. Few Examples: introduction line 52, results line 150

  1. we corrected grammatical errors in the manuscript.

-It may be easier to call “no remdesivir” arm as control arm in tables and figures. Figure 1 is key figure of paper. Unclear if the solid line on Kaplan Meier curve is control or Remdesivir arm? Based on data appears that solid line is control but it should be mentioned in figure legent.

R: we modified in Tables and Results section “no remdesivir” in “control group”. Moreover, we modified figure 1 title.

-Remdsivir was not given to patients with GFR < 30 OR ALT < 5 times upper limit normal. Which is based on current guidelines. This adds a selection bias to study. The control group may have more patients with kidney and liver disease who were not able to get Remdesivir. Can authors shed some light on how that selection bias was addressed.

R: this is another important observation. Our data showed that no statistically significant differences in kidney and liver diseases were reported in the 2 study groups. Then, these results allowed us to be able to select for propensity score matching the treatments, in order to reduce the bias of remdesivir use in different populations.

Round 2

Reviewer 1 Report

Authors sufficiently responded all comments.

No further comment.

Author Response

Thank you very much.

Reviewer 2 Report

No comments

Author Response

Thank you.

Reviewer 3 Report

I appreciate your efforts and thank you for addressing some of my comments from the prior review. I re-read your work and have the following concerns.

Major limitations:

The conclusion that mechanical ventilation is associated with 30- day mortality is based on 11 patients. Given such a low n this conclusion is not valid based on the data provided. The same is true for HFNC/NIV. Additionally, the Confidence interval of 0.954-336.73 with NIV further questions the validity of this conclusion.

The results of your study have clinical implications, but the manuscript is hard to read and understand because of sentence structure and writing style. I have cited a few examples below, but there are many more in the result and discussion section. I will recommend that authors present the results and discussions section to be easy to comprehend for the readers.

Line 18: “large population” 407 patients are not such a large cohort. There are multiple published studies on COVID with much larger N.

Give n and percentage of in-hospital mortality.

“Univariate analysis about patients treated…’ sentence needs grammatical correction.

Line 32 is grammatically incorrect.

Introduction

“evolving in a pandemic” it is a full-blown pandemic.

Line 44: delete then, and break the sentence in two for flow and readability

Line 48: especially is used twice—consider an alternate word

Line 52: "only remdesivir has been approved": there are several treatments approved for COVID-19, remdesivir is one of them

Line 62: objective statement needs simplification

Author Response

I appreciate your efforts and thank you for addressing some of my comments from the prior review. I re-read your work and have the following concerns.

Major limitations:

The conclusion that mechanical ventilation is associated with 30- day mortality is based on 11 patients. Given such a low n this conclusion is not valid based on the data provided. The same is true for HFNC/NIV. Additionally, the Confidence interval of 0.954-336.73 with NIV further questions the validity of this conclusion.

R: thank you for this important observation. The primary endpoint of our study was to evaluate the impact of remdesivir on outcomes of COVID-19 patients. We removed from the abstract informations about HFNC/NIV and mechanical ventilation and modified conclusions. In Table 3 the correct p-value is 0.054, we corrected it and deleted conclusions about this point. However, we did not consider HFNC/NIV and mechanical ventilation in the propensity score considering the small sample size (see also supplementary material). We reported this limitation in Discussion section.

The results of your study have clinical implications, but the manuscript is hard to read and understand because of sentence structure and writing style. I have cited a few examples below, but there are many more in the result and discussion section. I will recommend that authors present the results and discussions section to be easy to comprehend for the readers.

 R: Dear reviewer, to improve quality of the manuscript we modified it according with your suggestions but also used the English editing from MDPI. Then, now the manuscript has been corrected by a native English speaker.

Line 18: “large population” 407 patients are not such a large cohort. There are multiple published studies on COVID with much larger N.

Give n and percentage of in-hospital mortality.

“Univariate analysis about patients treated…’ sentence needs grammatical correction.

Line 32 is grammatically incorrect.

Introduction

“evolving in a pandemic” it is a full-blown pandemic.

Line 44: delete then, and break the sentence in two for flow and readability

Line 48: especially is used twice—consider an alternate word

Line 52: "only remdesivir has been approved": there are several treatments approved for COVID-19, remdesivir is one of them

Line 62: objective statement needs simplification

R: we modifed the manuscript as suggested.